# CLEAN: CURVATURE-AWARE MEMORY-EFFICIENT OPTIMIZER VIA NYSTRÖM SKETCHING

## ABSTRACT

Training large language models is constrained by a trade-off between optimizer memory and curvature information. While memory-saving optimizers often discard valuable second-order information, leading to slower convergence, full-matrix methods are prohibitively expensive. We introduce **CLEAN**, a curvature-aware and memory-efficient optimizer that resolves this dilemma. **CLEAN** approximates the left and right gradient covariances using randomized Nyström sketches, enabling balanced, two-sided preconditioning. The optimization proceeds by computing updates within a compact, low-rank subspace and then projecting them back to the full parameter space, capturing rich curvature information at a minimal memory cost. A key innovation in **CLEAN** is a projection-aware moment transport mechanism. As the low-rank subspace evolves, this transport realigns the optimizer's first and second moments to the new basis, which is critical for maintaining stability and avoiding performance degradation from stale statistics. **CLEAN**'s memory footprint is orders-of-magnitude smaller than Adam's, growing only linearly with the number of parameters. Our experiments show **CLEAN** is highly effective for fine-tuning, outperforming strong memory-efficient baselines, while also demonstrating competitive feasibility in pre-training scenarios.

## 1 INTRODUCTION

The rapid scaling of large language models (LLMs) has shifted the training bottleneck from model design to optimization. While larger models and longer sequences promise better performance, the optimizer often becomes the limiting factor. Maintaining first– and second–moment statistics can require as much or more memory than the model weights themselves, constraining the effective batch size and sequence length on commodity hardware. At the same time, methods that reduce this footprint typically discard curvature information, producing poorly conditioned updates and slowing convergence. Balancing memory efficiency with curvature fidelity has therefore emerged as a central challenge for efficient LLM training.

A range of optimizers have been proposed, each making a different trade–off between memory and curvature. *First–order methods* such as Adam (Kingma & Ba, 2014) and Adafactor (Shazeer & Stern, 2018) are lightweight, storing only diagonal statistics. Their low memory cost makes them practical for large models, but ignoring directional geometry often leads to slow convergence and higher sample complexity.. *Full–matrix second–order methods* such as Shampoo (Gupta et al., 2018) and KFAC (Martens & Grosse, 2015) explicitly capture curvature and can accelerate training, but their quadratic state and computation make them prohibitive at LLM scale. Morwani et al. (2024) investigated the theoretical underpinnings of Shampoo and suggested minor modifications—such as adopting a $1/2$ power instead of $1/4$—which are consistent with earlier empirical observations by (Anil et al., 2020). More recently, *low–rank projection methods* have emerged as a compromise. GaLore (Zhao et al., 2024) compresses gradients into a low–dimensional subspace, but it does not fully account for left– and right–side covariances and becomes unstable as the subspace drifts over time. LDAdam (Robert et al., 2024) adds projection–aware updates, yet still inherits GaLore's limitations. SOAP (Vyas et al., 2024) stabilizes low–frequency preconditioning by combining Shampoo's eigenbasis with Adam–style updates, but retains full covariance accumulators that remain costly in both memory and time. Despite these advances, no existing approach simultaneously achieves memory efficiency, curvature awareness, and long–horizon stability.

To fill these gaps, we introduce **CLEAN**, a curvature–aware yet memory–efficient optimizer that addresses these limitations. **CLEAN** is built on two key insights. First, the left and right gradient covariances contain most of the useful curvature but are effectively low–rank in practice. We therefore approximate them with randomized Nyström sketches, maintaining compact bases that reduce optimizer state from quadratic to linear in the layer dimensions. Second, when these bases evolve over time, exponential moving averages must be aligned with the new coordinates; otherwise the accumulated moments become inconsistent and updates unstable. To resolve this, **CLEAN** performs moment *transport* through change–of–basis transformations, ensuring that both first– and second–order statistics remain valid as the subspace shifts. The overall design is simple: each step forms a small *core update* in the sketched space, applies Adam's moment updates there, and then back–projects the result to the full parameter space. This procedure approximates balanced preconditioning while storing only $O((m+n)r + r^2)$ state per matrix layer—orders of magnitude smaller than full second–order methods. **CLEAN** is easy to implement, robust across a range of ranks and refresh periods, and achieves the speed of first–order methods with significantly reduced memory and improved sample efficiency.

Our contributions can be summarized as follows:

- We introduce **CLEAN**, a low–rank, projection–aware curvature optimizer that couples Nyström sketches of gradient covariances with Adam updates in a compact core, plus a principled mechanism for transporting moments across evolving subspaces.

- We analyze why this design approximates balanced preconditioning, provide memory/compute accounting, and propose practical stabilizations such as damping, eigenvalue clamping, and optional one–sided projections.

- We demonstrate **CLEAN**'s strong empirical performance in fine-tuning tasks, where it provides significant advantages over existing optimizers. We also demonstrate its effectiveness for pre-training of 130M parameter models, achieving competitive performance.

## 2 BACKGROUND AND PRELIMINARIES

**Adaptive optimization as preconditioning.** Most optimizers for deep learning can be understood as applying a preconditioner to the gradient. For a parameter vector $w_t$ with gradient $g_t = \nabla \phi_t(w_t)$, a generic update takes the form

$$w_{t+1} = w_t - \eta \, P_t^{-1/2} \, g_t,$$

where $P_t$ is a positive definite matrix encoding accumulated curvature information. Full–matrix AdaGrad (Duchi et al., 2011) sets $P_t = \sum_{s=1}^{t} g_s g_s^\top$, yielding well–conditioned updates, but requires $O(d^2)$ memory and inversion, which is infeasible for modern models with billions of parameters. Common approaches simplify $P_t$ to make optimization practical. *Diagonal preconditioners*, used in optimizers like Adam (Kingma & Ba, 2014), restrict $P_t$ to a diagonal matrix, capturing individual parameter scales but ignoring cross-parameter correlations. *One-sided preconditioners*, such as in GaLore (Zhao et al., 2024), apply transformations from either the left or right side, offering a compromise by capturing some structural information with less overhead than full-matrix methods.

**Balanced preconditioning for matrix layers.** Large models are composed of weight matrices $W \in \mathbb{R}^{m \times n}$. For such structured parameters, the full preconditioner is an $mn \times mn$ matrix. A practical surrogate is to maintain left and right covariance accumulators:

$$L_t = \sum_{s=1}^{t} G_s G_s^\top, \quad R_t = \sum_{s=1}^{t} G_s^\top G_s,$$

where $G_s$ is the gradient of $W$ at step $s$. The preconditioned gradient is then

$$\tilde{G}_t = L_t^{-1/2} \, G_t \, R_t^{-1/2},$$

which captures row and column geometry without constructing the full Kronecker product. This *balanced preconditioning* underlies Shampoo (Gupta et al., 2018), a popular *second-order preconditioning* method, and has become a standard approximation to full–matrix AdaGrad.

**Shampoo and practical variants.** Shampoo computes and stores per–layer $L_t$ and $R_t$ and applies fractional powers $L_t^{-\alpha}, R_t^{-\alpha}$. The original choice $\alpha = 1/4$ matches AdaGrad in theory, but later work found that $\alpha = 1/2$ often improves stability (Anil et al., 2020). However, fully storing $L_t$ and $R_t$ still incurs quadratic memory in layer width, which is prohibitive for very wide layers. Our work addresses this limitation by using a low-rank approximation.

**Projection–aware moment transport.** When low–rank bases $U_{t-1}$ are refreshed to $U_t$, stored moments $M_{t-1}, V_{t-1}$ no longer align with the new coordinates. Naively reusing them causes inconsistency and instability. A principled fix is to transport states via the change of basis:

$$M_t \leftarrow U_t^\top U_{t-1} M_{t-1} U_{t-1}^\top U_t,$$

(and similarly for $V_t$). This *projection–aware transport* preserves the semantics of exponential averaging and is critical when refreshes are infrequent.

## 3 ALGORITHM

In this section we present our memory-efficient, computationally light framework that applies low-rank compression to both the gradients and the optimizer state. The resulting algorithm carries out optimization in a reduced-dimensional subspace, preserving curvature information while sidestepping explicit second-order calculations. We start this section by introducing randomized Nyström approximation.

### 3.1 LOW-RANK APPROXIMATIONS WITH RANDOMIZED NYSTRÖM

Gradient covariances are often numerically low–rank: their spectrum decays quickly, with most energy concentrated in a small subspace. This motivates low–rank approximations. Given a PSD matrix $A \in \mathbb{R}^{m \times n}$ and a random (Gaussian) matrix $\Omega \in \mathbb{R}^{n \times r}$, the Nyström method builds

$$C = A\Omega, \quad W = \Omega^\top A \Omega, \quad \tilde{A} = CW^\dagger C^\top \simeq A. \tag{1}$$

The result is a rank–$r$ surrogate $\tilde{A}$ requiring only $O(nr + r^2)$ storage. Nyström sketches have strong theoretical guarantees (Gittens & Mahoney, 2016) and practical efficiency, making them attractive for optimizer states.

### 3.2 RANDOMIZED NYSTROM PRECONDITIONER

To address the high dimensionality and prohibitive memory requirements of full preconditioning, we leverage low-rank approximations of the gradient covariance. Specifically, we employ the randomized Nyström method (Gittens & Mahoney, 2016) to construct a memory-efficient preconditioner that captures essential curvature information. While the naive implementation in equation 1 can be numerically unstable, we adopt the stable and efficient implementation from (Tropp et al., 2017; Frangella et al., 2023), presented in Algorithm 2 in the Appendix.

Our approach approximates the full covariance matrices, $G^\top G$ and $GG^\top$, avoiding the computational expense of forming them explicitly. Let $\Omega_L \in \mathbb{R}^{m \times r}$ and $\Omega_R \in \mathbb{R}^{n \times r}$ be Gaussian random matrices. Following the procedure in Algorithm 2, we compute low-dimensional sketches $G^\top \Omega_L \in \mathbb{R}^{n \times r}$ and $G\Omega_R \in \mathbb{R}^{m \times r}$, followed by $G(G^\top \Omega_L) \in \mathbb{R}^{m \times r}$ and $G^\top(G\Omega_R) \in \mathbb{R}^{n \times r}$. This process yields the approximations:

$$L = U_L \Lambda_L U_L^\top \approx GG^\top, \quad R = U_R \Lambda_R U_R^\top \approx G^\top G. \tag{2}$$

Inspired by (Gupta et al., 2018), we construct the preconditioned gradient as

$$G^{\text{pre}} = L^{-1/2} G R^{-1/2} = (U_L \Lambda_L^{-1/2} U_L^\top) G (U_R \Lambda_R U_R^\top), \tag{3}$$

In our proposed Algorithm 1, we replace the raw gradient $G$ with its preconditioned counterpart $G^{\text{pre}}$. This modification leverages the spectral information encoded in $L$ and $R$, leading to better-conditioned updates and improved convergence behavior.

**Algorithm 1** CLEAN Optimizer

Preconditioner Approximations , Preconditioner Accumulations , Projection-aware Moments

Projected Gradient

---

1: **Input:** $W_t, G_t \in \mathbb{R}^{m \times n}$, learning rate $\eta$, betas $= (\beta_1, \beta_2)$, ranks $r_1 \leq m, r_2 \leq n$, accumulation weight $\mu$, scale factor $\alpha$, subspace update interval $T$, regularization factor $\rho$.
2: Initialize first-order moment $M_0 \in \mathbb{R}^{r_1 \times r_2} \leftarrow 0$
3: Initialize second-order moment $V_0 \in \mathbb{R}^{r_1 \times r_2} \leftarrow 0$
4: Initialize step $t \leftarrow 0$

---

5: $G_t \in \mathbb{R}^{m \times n} \leftarrow -\nabla_W \phi_t(W_t)$
6: **if** $t \mod T = 0$ **then**
7:     $U_{t,L}, \Lambda_{t,L} \leftarrow \text{RandNystromApprox}(G_t, \text{rank} = r_1)$        $\{L_t = U_{t,L}\Lambda_{t,L}U_{t,L}^\top \approx G_t G_t^\top\}$
8:     $U_{t,R}, \Lambda_{t,R} \leftarrow \text{RandNystromApprox}(G_t^\top, \text{rank} = r_2)$        $\{R_t = U_{t,R}\Lambda_{t,R}U_{t,R}^\top \approx G_t^\top G_t\}$
9:     **if** $t > 0$ **then**
10:          $[U_{t,L}, \Lambda_{t,L}, V_{t,L}] \leftarrow \text{SVD}\left(\left[\sqrt{\mu}\, U_{t-1,L}\Lambda_{t-1,L}^{1/2} \,\Big|\, \sqrt{1-\mu}\, U_{t,L}\Lambda_{t,L}^{1/2}\right], \text{rank} = r_1\right)$

              $[U_{t,R}, \Lambda_{t,R}, , V_{t,R}] \leftarrow \text{SVD}\left(\left[\sqrt{\mu}U_{t-1,R}\Lambda_{t-1,R}^{1/2} \,\Big|\, \sqrt{1-\mu}U_{t,R}\Lambda_{t,R}^{1/2}\right], \text{rank} = r_2\right)$

11: **else**
12:     $U_{t,L}, \Lambda_{t,L} \leftarrow U_{t-1,L}, \Lambda_{t-1,L}$        {Reuse the previous}
13:     $U_{t,R}, \Lambda_{t,R} \leftarrow U_{t-1,R}, \Lambda_{t-1,R}$        {Reuse the previous}
14: $S_t \leftarrow (\Lambda_{t,L} + \rho I)^{-1/2} U_{t,L}^\top G_t U_{t,R} (\Lambda_{t,R} + \rho I)^{-1/2} \in \mathbb{R}^{r_1 \times r_2}$
15: $G_t^{\text{pre}} \leftarrow U_{t,L} S_t U_{t,R}^\top$

---

16: **if** $t \mod T = 0$ and $t > 0$ **then**
17:
18:     $M_t \leftarrow \beta_1 \cdot U_{t,L}^\top U_{t-1,L} M_{t-1} U_{t-1,R}^\top U_{t,R} + (1 - \beta_1) \cdot S_t$

19:     $V_t \leftarrow \beta_2 \cdot \left( \begin{matrix} (1 - \beta_2^{t-1}) \cdot \Big|(U_{t,L}^\top U_{t-1,L})^2 (V_{t-1} - M_{t-1}^2)(U_{t-1,R}^\top U_{t,R})^2 \\ + (U_{t,L}^\top U_{t-1,L} M_{t-1} U_{t-1,R}^\top U_{t,R})^2\Big| \end{matrix} \right) + (1 - \beta_2) S_t^2$

20: **else**
21:     $M_t \leftarrow \beta_1 M_{t-1} + (1 - \beta_1) S_t$
22:     $V_t \leftarrow \beta_2 V_{t-1} + (1 - \beta_2) S_t^2$

23: $M_t \leftarrow M_t / (1 - \beta_1^t)$
24: $V_t \leftarrow V_t / (1 - \beta_2^t)$
25: $N_t \leftarrow M_t / (\sqrt{V_t} + \epsilon)$

---

26: $\tilde{G}_t^{\text{pre}} \leftarrow U_{t,L} N_t U_{t,R}^\top$        {Project back to original space}
27: $W_t \leftarrow W_{t-1} + \alpha\eta \cdot \tilde{G}_t^{\text{pre}}$
28: $t \leftarrow t + 1$
29: **return** $W_t$

---

### 3.3 THE CLEAN OPTIMIZER

We now give a detailed description of the **CLEAN** optimizer. Pseudo-code is provided in Algorithm 1; here we elaborate on each step and the design choices that enable an efficient, low-memory adaptive preconditioner.

**Preliminaries.** At iteration $t$ we denote the (negated) gradient by $G_t \in \mathbb{R}^{m \times n}$, i.e. $G_t = -\nabla_W \phi_t(W_t)$. **CLEAN** maintains two low-rank, positive semi-definite (PSD) preconditioners:

a left preconditioner $L_t$ (an approximation of $G_t G_t^\mathsf{T}$) with rank $r_1 \ll m$ and a right preconditioner $R_t$ (an approximation of $G_t^\mathsf{T} G_t$) with rank $r_2 \ll n$. We never explicitly form the full $m \times m$ or $n \times n$ matrices $L_t$ and $R_t$. Instead, using the randomized Nyström routine (Algorithm 2), we obtain compact factors

$$U_{t,L} \in \mathbb{R}^{m \times r_1}, \quad \Lambda_{t,L} \in \mathbb{R}^{r_1 \times r_1}, \qquad U_{t,R} \in \mathbb{R}^{n \times r_2}, \quad \Lambda_{t,R} \in \mathbb{R}^{r_2 \times r_2}, \tag{4}$$

such that $L_t \approx U_{t,L} \Lambda_{t,L} U_{t,L}^\mathsf{T}$ and $R_t \approx U_{t,R} \Lambda_{t,R} U_{t,R}^\mathsf{T}$. Subspace updates are performed every $T$ iterations; between updates the previous factors are reused.

**Smoothing / accumulation of preconditioners.** To avoid abrupt changes in the low-rank basis and to provide temporal smoothing of the preconditioner, **CLEAN** accumulates the previous and current Nyström estimates. A natural additive accumulation is

$$\widetilde{L}_t = \mu L_{t-1} + (1-\mu)L_t, \qquad \widetilde{R}_t = \mu R_{t-1} + (1-\mu)R_t, \tag{5}$$

with smoothing weight $\mu \in [0, 1]$. Since each $L$ is PSD, there exist Cholesky-like factors $C_{t-1,L}$ and $C_{t,L}$ satisfying $L_{t-1} = C_{t-1,L} C_{t-1,L}^\mathsf{T}$ and $L_t = C_{t,L} C_{t,L}^\mathsf{T}$, where

$$C_{t-1,L} = U_{t-1,L} \Lambda_{t-1,L}^{1/2}, \qquad C_{t,L} = U_{t,L} \Lambda_{t,L}^{1/2}. \tag{6}$$

We then form the concatenated factor

$$K_L = \left[ \sqrt{\mu}\, C_{t-1,L} \mid \sqrt{1-\mu}\, C_{t,L} \right] \tag{7}$$

so that $\widetilde{L}_t = K_L K_L^\mathsf{T}$. A truncated SVD (or economy SVD) of $K_L$ yields a rank-$r_1$ factorization $\widetilde{L}_t \approx U_{t,L} \Lambda_{t,L} U_{t,L}^\mathsf{T}$ without ever forming full $m \times m$ matrices. The same procedure is applied to produce $\widetilde{R}_t$ from the right factors. In the pseudocode this concatenation plus truncation is implemented in line 10; when $t$ is not a refresh step the algorithm simply reuses the previous factors.

**Preconditioning in the low-rank subspace.** Following Gupta et al. (2018), we define the preconditioned gradient

$$G_t^{\mathrm{pre}} \triangleq L_t^{-1/2} G_t R_t^{-1/2}. \tag{8}$$

Using the Nyström factors this becomes

$$G_t^{\mathrm{pre}} = (U_{t,L} \Lambda_{t,L}^{-1/2} U_{t,L}^\mathsf{T})\, G_t\, (U_{t,R} \Lambda_{t,R}^{-1/2} U_{t,R}^\mathsf{T}). \tag{9}$$

Forming $G_t^{\mathrm{pre}}$ explicitly would require $O(mn)$ memory and computation. Instead, we compute the small, projected matrix

$$S_t = \Lambda_{t,L}^{-1/2} U_{t,L}^\mathsf{T} G_t U_{t,R} \Lambda_{t,R}^{-1/2} \quad \in \mathbb{R}^{r_1 \times r_2}, \tag{10}$$

which represents $G_t^{\mathrm{pre}}$ in the low-dimensional coordinate system defined by $U_{t,L}$ and $U_{t,R}$.

**Projection-aware adaptive moments.** **CLEAN** maintains Adam-style first and second moments in the compact projected space (lines 18–27 of Algorithm 1). Let $M_t$ and $V_t$ denote the first- and second-moment estimates in $\mathbb{R}^{r_1 \times r_2}$ space. Inspired by LDAdam (Robert et al., 2024), on subspace-refresh steps (when the bases $U_{t,*}$ have changed) we perform *projection-aware* moment accumulation: the previous moments are transported into the new bases using the alignment matrices $U_{t,*}^\top U_{t-1,*}$. Concretely, when $t$ is a refresh step we set

$$M_t \leftarrow \beta_1 U_{t,L}^\top U_{t-1,L} M_{t-1} U_{t-1,R}^\top U_{t,R} + (1-\beta_1) S_t, \tag{11}$$

and a compatible expression is used for $V_t$ to account for second moments and squared terms (see Algorithm 1 for the exact form). For non-refresh steps we use the usual exponential moving averages

$$M_t \leftarrow \beta_1 M_{t-1} + (1-\beta_1) S_t, \qquad V_t \leftarrow \beta_2 V_{t-1} + (1-\beta_2) S_t^2. \tag{12}$$

Bias correction is applied to both moments by dividing by $1 - \beta_1^t$ and $1 - \beta_2^t$, respectively. The normalized projected update is then $N_t = \frac{M_t}{\sqrt{V_t} + \epsilon}$, computed elementwise.

**Projection back and parameter update.** The small matrix $N_t \in \mathbb{R}^{r_1 \times r_2}$ is re-expanded to the original parameter space via $\widetilde{G}_t^{\text{pre}} = U_{t,L} N_t U_{t,R}^{\mathsf{T}}$, which is an $m \times n$ matrix but is obtained by low-rank factors. The parameter update (line 30) is $W_t \leftarrow W_{t-1} + \alpha\eta \cdot \widetilde{G}_t^{\text{pre}}$, where $\eta$ is the base learning rate and $\alpha$ is a scale factor.

## 4 THEORETICAL RESULTS

In this section, we analyze the projection-aware property of **CLEAN**, its convergence and discuss its computational advantages.

**Projection-aware optimizer** As introduced in Section 3.3 and Algorithm 1, **CLEAN** adopts projection-aware update rules inspired by LDAdam (Robert et al., 2024). This ensures that when the projection subspace is refreshed, the optimizer's historical states (first and second moments) are correctly transported to the new basis. This prevents instability and preserves the accumulated information. The detailed derivation of the projection-aware moment update rules can be found in Appendix A.

**Convergence** The following theorem establishes the convergence of **CLEAN**.

**Theorem 4.1.** *(Convergence of CLEAN with fixed projections). Suppose that the gradient has the parametric form $G_t = \sum_{i=1}^N A_i - \sum_{i=1}^N B_i W_t C_i$ where $N$ is a batch size and the functions $A_i$, $B_i$ and $C_i$ have $L_A, L_B$ and $L_C$ continuity, respectively with respect to $W$. Let $\|W\|_F \leq M$ with $M$ constant. Also, assume that the projection matrices remain constant during training. Then for a suitable learning rate $\eta$, the **CLEAN** optimizer satisfies*

$$\|S_t\|_F \leq \left[1 - \frac{\eta}{\lambda_0}(\kappa_{t-1} - C)\right] \|S_{t-1}\|_F,$$

*where $S_t$ is the projected gradient, $\lambda_0$ is the minimum eigenvalue of the preconditioner approximation, $\kappa_{t-1}$ is a curvature term, and $C$ is a constant related to the Lipschitz constants. A detailed statement of the theorem and its proof are provided in Appendix A.*

**Costs and advantages.** **CLEAN** avoids the $O(mn)$ memory and computation that would be required to form full preconditioned gradients by (i) representing preconditioners with Nyström factors $U_{t,*}, \Lambda_{t,*}$, (ii) operating on the small projected matrix $S_t \in \mathbb{R}^{r_1 \times r_2}$, and (iii) using projection-aware moment transport when bases change. Memory is therefore dominated by the factors $U_{t,L}, U_{t,R}$ and the small moments $M_t, V_t$, i.e. $O(mr_1 + nr_2 + r_1 r_2)$, and per-iteration work is similarly reduced relative to full-matrix adaptive methods. Smoothing via concatenation-and-truncation ensures the accumulated preconditioner remains PSD and results in gradual basis evolution, which improves stability when subspaces are refreshed periodically. Table 1 compares the memory for CLEAN vs Adam, LoRA, Galore and SOAP. For **CLEAN** we assume $r_1 = r_2 = r$. This highlights that **CLEAN** is a lightweight optimization method that achieves reduced memory overhead

Table 1: Comparison of optimizer states and weights memory for different baselines

|  | Adam | LoRA | GaLore | SOAP | CLEAN |
|---|---|---|---|---|---|
| **Weight** | $mn$ | $mn + mr + nr$ | $mn$ | $mn$ | $mn$ |
| **Optim. States** | $2mn$ | $2mr + 2nr$ | $mr + 2nr$ | $2m^2 + 2n^2 + 2mn$ | $mr + nr + 2r^2$ |

## 5 RELATED WORK

**Low-rank Projection Methods.** A popular approach to reducing optimizer memory is to project gradients onto a low-rank subspace. GaLore (Zhao et al., 2024) introduced this strategy by performing gradient updates within a low-rank subspace defined by the singular value decomposition (SVD) of recent gradients. While effective at saving memory, this one-sided projection can be unstable as the subspace drifts over time. LDAdam (Robert et al., 2024) and Subtrack++ (Rajabi et al., 2025) attempted to mitigate this by incorporating projection-aware updates and more robust subspace

tracking mechanisms. However, these methods still primarily operate on one-sided approximations and do not fully capture the row-column geometry of the gradient covariances.

**Second-order Preconditioning.** Another line of work focuses on capturing gradient curvature information using second-order methods. K-FAC (Martens & Grosse, 2015) approximates the Fisher information matrix with Kronecker products, while Shampoo (Gupta et al., 2018) directly approximates the full-matrix AdaGrad preconditioner by maintaining separate row and column preconditioners. These balanced, two-sided preconditioners are powerful but require storing full-sized covariance matrices, making them memory- and compute-intensive. Recent work like SOAP (Vyas et al., 2024) combines Shampoo's preconditioning with first-order updates but still retains the costly full-matrix accumulators.

**Positioning Our Method.** Our optimizer, **CLEAN**, bridges the gap between these two approaches. It combines the memory efficiency of low-rank methods with the curvature-awareness of two-sided preconditioning by using randomized Nyström sketches, while ensuring stability through projection-aware moment transport.

## 6 EXPERIMENTS

We evaluate **CLEAN** on both **fine-tuning** and a small-scale **pre-training feasibility study**. Baselines include AdamW (Kingma & Ba, 2014), SOAP (Vyas et al., 2024), and GaLore (Zhao et al., 2024). To demonstrate memory efficiency relative to parameter-efficient fine-tuning (PEFT), we also include LoRA (Hu et al., 2021). Like GaLore and other PEFT approaches, our method restricts low-rank compression to the two-dimensional matrices in self-attention layers, while larger embedding and output layers are optimized with standard Adam. To ensure reproducibility, we include the complete set of hyperparameters in the Appendix D. All experiments are conducted on NVIDIA H100 GPUs.

### 6.1 FINE-TUNING ON GLUE

To evaluate the fine-tuning performance of **CLEAN**, we test it on the GLUE benchmark (Wang et al., 2018) using the RoBERTa-base model (Liu et al., 2019). We compare against several strong baselines, including AdamW, LoRA, GaLore, and SOAP. Due to time limitations, we trained for 3 epochs with a batch size of 16, using rank-$r = 32$ for **CLEAN** and rank-$r = 4, 8$ for GaLore and LoRA to ensure comparable memory usage. We also tune over a set of learning rates reported in Table 4. Table 2 reports the best average over 3 seeds. As shown in this table **CLEAN** achieves competitive performance, particularly on the STSB task, close to the full-rank SOAP optimizer. While AdamW and SOAP act as full-rank baselines, **CLEAN** demonstrates competitive average performance compared to low-rank baselines, highlighting its potential as an effective and memory-efficient fine-tuning optimizer. As shown in Table 2, **CLEAN** has the lowest memory among all the optimizers while achieving fast runtime with the highest average accuracy among low-rank methods.

Table 2: GLUE Fine-tuning Results (RoBERTa-base). Results are percentages ($\times 100$). Best results are in **bold**. Results are averaged across 3 seeds and best across 5 learning rates.

| Method | MRPC | STS-B | CoLA | SST-2 | QNLI | QQP | MNLI | Average | Time | Optim. State (MB) |
|---|---|---|---|---|---|---|---|---|---|---|
| Training Samples | 3.7k | 7k | 8.5k | 67k | 105k | 364k | 393k | - | - | - |
| AdamW | 88.97 | 90.21 | 59.94 | 94.23 | 92.73 | 91.42 | 87.77 | 86.47 | 1.00 | 951.0 |
| SOAP (T=32) | 89.05 | 89.95 | 57.55 | 92.55 | 91.76 | 90.82 | 85.01 | 85.24 | 4.56 | 2853.0 |
| GaLore (T=500, r=4) | 86.27 | 88.34 | 55.03 | 93.50 | 91.72 | 89.66 | 86.08 | 84.37 | 1.13 | 301.9 |
| GaLore (T=500, r=8) | **87.58** | 88.52 | 54.84 | **93.54** | **92.06** | 89.80 | **86.46** | 84.68 | 1.14 | 305.3 |
| LoRA (r=4) | 85.78 | 88.56 | **57.27** | 93.46 | 89.52 | 89.76 | 84.26 | 84.09 | 0.85 | 308.0 |
| LoRA (r=8) | 86.68 | 88.84 | 55.94 | 93.23 | 89.60 | **89.81** | 84.39 | 84.07 | **0.84** | 313.1 |
| **CLEAN** (T=32, r=32) | 87.25 | **89.96** | 56.89 | 93.20 | 91.63 | 89.59 | 85.75 | **84.90** | 1.09 | **299.1** |

### 6.2 ABLATION STUDIES

We performed an ablation study to assess the individual and combined contributions of different hyperparameters on **CLEAN** optimizer.

**Rank sensitivity.** We vary rank $r \in \{4, 8, 16, 32, 64\}$ while fixing refresh $T = 32$. Table 3 shows that performance is stable across ranks: accuracy differences are less than $1\%$ on the average performance, while runtime grows modestly with rank. This indicates **CLEAN** is robust to rank choice and practical even at small $r$.

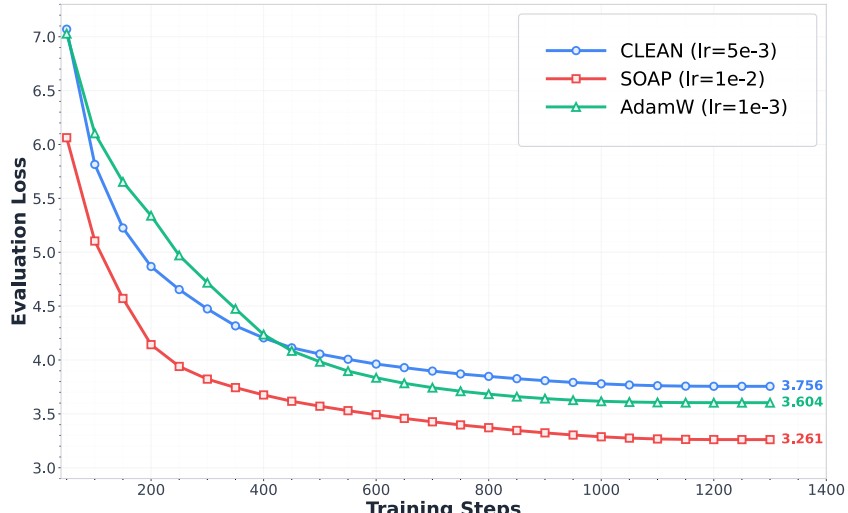

Figure 1: Pre-training evaluation loss vs training step for of Llama 150M on C4 dataset

**Subspace Update Interval.** We vary the subspace update interval $T \in 3, 10, 32, 100, 200, 500$ while keeping the rank fixed at $r = 32$. As reported in Table 3, accuracy peaks slightly at $T = 100$, whereas runtime follows the expected amortization trend: increasing $T$ accelerates **CLEAN**, reaching AdamW's speed at $T = 500$. These results highlight both the low sensitivity of **CLEAN** to $T$, subspace update interval, and its ability to efficiently balance accuracy and computational cost.

**Preconditioner accumulation weight $\mu$.** We test $\mu \in \{0.05, 0.1, 0.3, 0.5, 0.75\}$. $\mu = 0.5$ has the highest accuracy, while highlighting low sensitivity of **CLEAN** optimizer to preconditioner accumulation weight $\mu$ Table 3 summarizes.

**Projection-aware transport.** Disabling projection-aware transport degrades accuracy ($\sim$ more than a point on GLUE). This confirms transport is critical for stable training under subspace update.

### 6.3 PRE-TRAINING FEASIBILITY STUDY

**Setup.** We evaluate the pre-training task on a subset of C4 dataset (Raffel et al., 2020). Due to time and computational constraints, our experiments focus on a smaller Llama-family Touvron et al. (2023) model with 130M parameters. Following the Chinchilla scaling rule (Hoffmann et al., 2022), we allocate 20 training tokens per model parameter. Training is conducted with a token batch size of $\approx 2M$ and a sequence length of 1024, corresponding to 1300 steps. Training uses bf16 precision. We compare **CLEAN** with AdamW, SOAP. We perform a hyperparameter sweep over learning rates $\{5e\text{-}4, 1e\text{-}3, 5e\text{-}3, 1e\text{-}2\}$, reporting the best results for ranks $r = 256$ for **CLEAN** in Figure 1.

**Results.** We benchmark our method with AdamW and SOAP to show that, even within a low-rank subspace, it can compete with full-rank optimizers with less memory. Figure 1 shows evaluation loss curves. This demonstrates feasibility for pretraining while highlighting directions to close the gap (dynamic rank allocation, refresh scheduling, and hyperparameter tuning).

### 6.4 DISCUSSION

Our experiments highlight three main findings:

Table 3: CLEAN Ablation Study (RoBERTa-base). Results are percentages ($\times 100$). Best results are in **bold**. Results are averaged across 3 seeds and the best across 5 learning rates.

| Configuration | MRPC | STS-B | CoLA | SST-2 | QNLI | QQP | MNLI | Average | Time | Optim. State (MB) |
|---|---|---|---|---|---|---|---|---|---|---|
| *Rank Variations (T=32, $\mu$=0.1, s=1):* | | | | | | | | | | |
| **CLEAN** (r=4) | 86.27 | 88.29 | 55.16 | 92.70 | 89.75 | 89.30 | 84.43 | 83.70 | 1.03 | 298.5 |
| **CLEAN** (r=8) | 87.99 | 88.94 | 54.85 | 93.04 | 90.19 | 89.02 | 84.96 | 84.14 | 1.04 | 298.5 |
| **CLEAN** (r=16) | 87.66 | 89.39 | 54.46 | 92.89 | 91.04 | 89.05 | 85.21 | 84.24 | 1.06 | 298.6 |
| **CLEAN** (r=32) | 87.25 | 89.81 | 53.28 | 93.12 | 91.60 | 89.21 | 85.10 | 84.20 | 1.09 | 299.1 |
| **CLEAN (r=64)** | **88.24** | **89.94** | **56.38** | **93.08** | **91.70** | **88.93** | **84.44** | **84.67** | **1.18** | **300.8** |
| *Subspace Update Interval Variations (r=32, $\mu$=0.1, s=1):* | | | | | | | | | | |
| **CLEAN** (T=3) | 86.85 | 89.26 | 52.15 | 91.90 | 89.61 | 87.50 | 83.60 | 82.98 | 1.94 | 299.1 |
| **CLEAN** (T=10) | 87.17 | 89.90 | 53.39 | 92.32 | 90.51 | 88.35 | 84.31 | 83.71 | 1.29 | 299.1 |
| **CLEAN** (T=32) | 87.25 | 89.81 | 53.28 | 93.12 | 91.60 | 89.21 | 85.10 | 84.20 | 1.09 | 299.1 |
| **CLEAN (T=100)** | **87.99** | **89.62** | **56.15** | **93.12** | **91.49** | **89.70** | **85.83** | **84.84** | **1.03** | **299.1** |
| **CLEAN** (T=200) | 87.25 | 89.50 | 55.44 | 93.54 | 91.57 | 89.77 | 85.97 | 84.72 | 1.01 | 299.1 |
| **CLEAN** (T=500) | 87.75 | 88.81 | 56.03 | 93.43 | 91.18 | 89.90 | 85.96 | 84.72 | 1.00 | 299.1 |
| *Accumulation Weight Variations (T=32, r=32, s=1):* | | | | | | | | | | |
| **CLEAN** ($\mu$=0.05) | 87.09 | 89.63 | 54.60 | 92.74 | 91.47 | 89.19 | 85.01 | 84.25 | 1.10 | 299.1 |
| **CLEAN** ($\mu$=0.1) | 87.25 | 89.81 | 53.28 | 93.12 | 91.60 | 89.21 | 85.10 | 84.20 | 1.09 | 299.1 |
| **CLEAN** ($\mu$=0.3) | 87.83 | 89.85 | 56.25 | 93.27 | 91.52 | 89.35 | 85.50 | 84.79 | 1.09 | 299.1 |
| **CLEAN ($\mu$=0.5)** | **87.25** | **89.96** | **56.89** | **93.20** | **91.63** | **89.59** | **85.75** | **84.90** | **1.09** | **299.1** |
| **CLEAN** ($\mu$=0.75) | 88.15 | 89.79 | 55.39 | 93.20 | 91.48 | 89.75 | 86.01 | 84.82 | 1.08 | 299.1 |
| *Additional Configurations (T=100, r=32, $\mu$=0.95, s=1):* | | | | | | | | | | |
| **CLEAN** (proj-off) | 85.78 | 88.60 | 53.95 | 92.74 | 90.13 | N/A | 85.71 | 82.82 | 1.03 | 299.1 |
| **CLEAN (proj-on)** | **87.66** | **89.74** | **55.76** | **93.08** | **91.41** | **N/A** | **86.13** | **83.97** | **1.03** | **299.1** |

- **Fine-tuning.** On GLUE, **CLEAN** achieves accuracy comparable to AdamW and SOAP while using drastically less optimizer state memory. Compared to PEFT baselines such as LoRA and GaLore, **CLEAN** delivers higher overall accuracy at smaller memory budgets.

- **Robustness.** Ablation studies show that accuracy is insensitive to rank, subspace update interval and preconditioner accumulation weights within a broad range, making **CLEAN** easy to deploy. Projection-aware transport is essential: disabling it reduces accuracy by more than 1 point.

- **Pre-training feasibility.** In small-scale pre-training, **CLEAN** competes with AdamW and SOAP. This gap highlights the need for adaptive rank allocation, subspace update interval, and hyperparameter tuning, which we leave to future work.

Overall, **CLEAN** provides a practical optimizer for memory-constrained fine-tuning with clear feasibility for pre-training. It achieves a favorable balance of accuracy, efficiency, and memory usage, making it suitable for single-accelerator training scenarios.

# 7 CONCLUSION

In this work, we introduced **CLEAN**, a memory-efficient optimizer that retains curvature information by coupling randomized Nyström sketches with projection-aware moment transport. Our experiments demonstrate that **CLEAN** achieves a compelling trade-off between memory efficiency and performance. It is particularly effective for fine-tuning language models, where it delivers competitive accuracy with a substantially smaller memory footprint than traditional optimizers. The feasibility study on pre-training also highlights its potential for training large models from scratch under constrained hardware. As part of future work, we aim to investigate potential enhancements to the design of **CLEAN**, specifically focusing on proposing its general form for tensor structure gradients, making it applicable to tensors of arbitrary dimensionality.

## REPRODUCIBILITY STATEMENT

To ensure reproducibility, we provide detailed descriptions of our model architecture, training procedures, and evaluation metrics in the main text and Appendix. Additionally, all experiments, including baseline comparisons and ablation studies, are documented with sufficient detail to allow independent replication. We also release the code and scripts to reproduce all results at `https://github.com/anon-code-2025/code_for_paper_submission/blob/main/clean.py`.

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
