## A USE OF LARGE LANGUAGE MODELS (LLMS)

In preparing this manuscript, we employed a Large Language Model (LLM) as a general-purpose writing assistant. Specifically, the LLM was used to polish the language, improve clarity and flow, and enhance the presentation of the text. All technical content, experimental design, data analysis, and model development were performed independently by the authors. The LLM was not used to generate any novel scientific ideas, experimental results, or interpretations.

## B DERIVATION OF PROJECTION-AWARE MOMENT UPDATES

The projection-aware update rule for the first-moment estimate arises from the multiplications with the matrices $U_{t,L}^{\mathsf{T}} U_{t-1,L} \in \mathbb{R}^{r_1 \times r_1}$ and $U_{t-1,R}^{\mathsf{T}} U_{t,R} \in \mathbb{R}^{r_2 \times r_2}$. This enables an efficient shift between subspaces without generating any intermediate high-dimensional matrices. Since both matrices $U_{t,L}^{\mathsf{T}} U_{t-1,L}$ and $U_{t-1,R}^{\mathsf{T}} U_{t,R}$ are orthogonal, they represent the change of basis between the two subspaces. Let

$$B_{t-1,L} = \begin{bmatrix} b_{t-1,L}^1, & \dots, & b_{t-1,L}^{r_1} \end{bmatrix} \quad \text{and} \quad B_{t,L} = \begin{bmatrix} b_{t,L}^1, & \dots, & b_{t,L}^{r_1} \end{bmatrix},$$

denote orthonormal bases for the subspaces $U_{t-1,L}$ and $U_{t,L}$ at time steps $t-1$ and $t$, respectively. Similarly, let

$$B_{t-1,R} = \begin{bmatrix} b_{t-1,R}^1, & \dots, & b_{t-1,R}^{r_2} \end{bmatrix} \quad \text{and} \quad B_{t,R} = \begin{bmatrix} b_{t,R}^1, & \dots, & b_{t,R}^{r_2} \end{bmatrix},$$

denote orthonormal bases for the subspaces $U_{t-1,R}$ and $U_{t,R}$ at time steps $t-1$ and $t$, respectively. Let $A_{t-1} \in \mathbb{R}^{r_1 \times r_2}$ and we want to change its basis to $A_t = U_{t,L}^{\mathsf{T}} U_{t-1,L} A_{t-1} U_{t-1}^{\mathsf{T}} U_{t,R}$, then the $p$-th row and the $q$-th column of a matrix $A_{t-1}$ transforms under the change of basis to time step $t$ is

$$A_t^{ij} = \sum_{p=1}^{r_1} \sum_{q=1}^{r_2} \langle b_{t,L}^i, b_{t-1,L}^p \rangle A_{t-1}^{pq} \langle b_{t-1,R}^q, b_{t,R}^j \rangle, \tag{13}$$

where we use superscripts to denote the elements of a matrix. Followed by analysis by (Robert et al., 2024), the first and second moments in Adam represent the exponentially time–weighted expectation at time $t$ with decay parameter $\beta$, i.e., $M_t = \mathbb{E}_{t,\beta_1}[S_t]$ and $V_t = \mathbb{E}_{t,\beta_2}[S_t^2]$. The first-moment estimate can be expressed under a change of basis through the transformation matrices $U_{t,L}^{\mathsf{T}} U_{t-1,L}$ and $U_{t-1,R}^{\mathsf{T}} U_{t,R}$. In particular, if $b_{t,L}^i$ denote the $i$-th row basis vector at step $t$, then the $ij$-th entry of the matrix $S_t$ at time step $t$ is denoted by $\langle b_{t,L}^i, S_t b_{t,R}^j \rangle$, when it has $B_{t,L}$ and $B_{t,R}$ as left and right subspaces at time $t$. With this notation and eqn. equation 13

$$M_t^{ij} = \mathbb{E}_{t,\beta_1}[S_t^{ij}] = \mathbb{E}_{t,\beta_1}[\langle b_{t,L}^i, S_t b_{t,R}^j \rangle]$$
$$= \sum_{p=1}^{r_1} \sum_{q=1}^{r_2} \langle b_{t,L}^i, b_{t-1,L}^p \rangle \mathbb{E}_{t,\beta_1}[\langle b_{t-1,L}^p, S_t b_{t-1,R}^q \rangle] \langle b_{t-1,R}^q, b_{t,R}^j \rangle$$
$$= (U_{t,L}^{\mathsf{T}} U_{t-1,L} M_{t-1} U_{t-1}^{\mathsf{T}} U_{t,R})^{ij}.$$

With the same approach, we can change the basis for the second moment as well:

$$V_t^{ij} = \mathbb{E}_{t,\beta_2}[(S_t^{ij})^2] = \mathbb{E}_{t,\beta_2}[\langle b_{t,L}^i, S_t b_{t,R}^j\rangle^2]$$

$$= \left(\sum_{p=1}^{r_1}\sum_{q=1}^{r_2}\langle b_{t,L}^i, b_{t-1,L}^p\rangle \mathbb{E}_{t,\beta_2}[\langle b_{t-1,L}^p, S_t b_{t-1,R}^q\rangle]\langle b_{t-1,R}^q, b_{t,R}^j\rangle\right)^2$$

$$= \sum_{p=1}^{r_1}\sum_{q=1}^{r_2}\langle b_{t,L}^i, b_{t-1,L}^p\rangle^2 \mathbb{E}_{t,\beta_2}[\langle b_{t-1,L}^p, S_t b_{t-1,R}^q\rangle^2]\langle b_{t-1,R}^q, b_{t,R}^j\rangle^2$$

$$+ \sum_{k\neq l}\sum_{k'\neq l'}\langle b_{t,L}^i, b_{t-1,L}^k\rangle \mathbb{E}_{t,\beta_2}[\langle b_{t-1,L}^k, S_t b_{t-1,R}^{k'}\rangle]\langle b_{t-1,R}^{k'}, b_{t,R}^j\rangle$$

$$\langle b_{t,L}^i, b_{t-1,L}^l\rangle \mathbb{E}_{t,\beta_2}[\langle b_{t-1,L}^l, S_t b_{t-1,R}^{l'}\rangle]\langle b_{t-1,R}^{l'}, b_{t,R}^j\rangle$$

$$= \sum_{p=1}^{r_1}\sum_{q=1}^{r_2}\langle b_{t,L}^i, b_{t-1,L}^p\rangle^2 V_{t-1}^{pq}\langle b_{t-1,R}^q, b_{t,R}^j\rangle^2$$

$$+ \sum_{k\neq l}\sum_{k'\neq l'}\langle b_{t,L}^i, b_{t-1,L}^k\rangle M_{t-1}^{k,k'}\langle b_{t-1,L}^{k'}, b_{t-1,L}^j\rangle$$

$$\langle b_{t,L}^i, b_{t-1,L}^l\rangle M_{t-1}^{l,l'}\langle b_{t-1,R}^{l'}, b_{t,R}^j\rangle. \tag{14}$$

We employ the following equation to rewrite the second term in the last equality:

$$\sum_{k,l}\sum_{k',l'}\langle b_{t,L}^i, b_{t-1,L}^k\rangle M_{t-1}^{k,k'}\langle b_{t-1,L}^{k'}, b_{t-1,L}^j\rangle\langle b_{t,L}^i, b_{t-1,L}^l\rangle M_{t-1}^{l,l'}\langle b_{t-1,R}^{l'}, b_{t,R}^j\rangle$$

$$= \sum_{k,k'}\langle b_{t,L}^i, b_{t-1,L}^k\rangle^2 (M_{t-1}^{k,k'})^2\langle b_{t-1,L}^{k'}, b_{t-1,L}^j\rangle^2 +$$

$$\sum_{k\neq l}^{r_1}\sum_{k'\neq l'}^{r_2}\langle b_{t,L}^i, b_{t-1,L}^k\rangle M_{t-1}^{k,k'}\langle b_{t-1,L}^{k'}, b_{t-1,L}^j\rangle\langle b_{t,L}^i, b_{t-1,L}^l\rangle M_{t-1}^{l,l'}\langle b_{t-1,R}^{l'}, b_{t,R}^j\rangle.$$

With this, we can rewrite eqn. equation 14:

$$V_t^{ij} = \mathbb{E}_{t,\beta_2}[(S_t^{ij})^2] = \sum_{p=1}^{r_1}\sum_{q=1}^{r_2}\langle b_{t,L}^i, b_{t-1,L}^p\rangle^2 V_{t-1}^{pq}\langle b_{t-1,R}^q, b_{t,R}^j\rangle^2$$

$$+ \sum_{k,l}\sum_{k',l'}\langle b_{t,L}^i, b_{t-1,L}^k\rangle M_{t-1}^{k,k'}\langle b_{t-1,L}^{k'}, b_{t-1,L}^j\rangle\langle b_{t,L}^i, b_{t-1,L}^l\rangle M_{t-1}^{l,l'}\langle b_{t-1,R}^{l'}, b_{t,R}^j\rangle$$

$$= \sum_{p=1}^{r_1}\sum_{q=1}^{r_2}\langle b_{t,L}^i, b_{t-1,L}^p\rangle^2 V_{t-1}^{pq}\langle b_{t-1,R}^q, b_{t,R}^j\rangle^2$$

$$+ \sum_{k,l}\sum_{k',l'}\langle b_{t,L}^i, b_{t-1,L}^k\rangle M_{t-1}^{k,k'}\langle b_{t-1,L}^{k'}, b_{t-1,L}^j\rangle\langle b_{t,L}^i, b_{t-1,L}^l\rangle M_{t-1}^{l,l'}\langle b_{t-1,R}^{l'}, b_{t,R}^j\rangle$$

$$- \sum_{k,k'}\langle b_{t,L}^i, b_{t-1,L}^k\rangle^2 (M_{t-1}^{k,k'})^2\langle b_{t-1,R}^{k'}, b_{t,R}^j\rangle^2$$

$$= \sum_{p=1}^{r_1}\sum_{q=1}^{r_2}\langle b_{t,L}^i, b_{t-1,L}^p\rangle^2 (V_{t-1}^{pq} - (M_{t-1}^{pq})^2)\langle b_{t-1,R}^q, b_{t,R}^j\rangle^2$$

$$+ \left(\langle b_{t,L}^i, b_{t-1,L}^p\rangle M_{t-1}^{p,q}\langle b_{t-1,L}^q, b_{t-1,L}^j\rangle\right)^2$$

$$= (U_{t,L}^\mathsf{T}U_{t-1,L})^2(V_{t-1} - M_{t-1}^2)(U_{t-1,R}^\mathsf{T}U_{t,R})^2 + (U_{t,L}^\mathsf{T}U_{t-1,L}M_{t-1}U_{t-1,R}^\mathsf{T}U_{t,R})^2.$$

## C  CONVERGENCE ANALYSIS

**Theorem C.1.** *(**Convergence of Clean with fixed projections**). Suppose that the gradient has the parametric form $G_t = \sum_{i=1}^N A_i - \sum_{i=1}^N B_i W_t C_i$ where $N$ is a batch size and the func-*

*tions $A_i, B_i$ and $C_i$ have $L_A, L_B$ and $L_C$ continuity, respectively with respect to $W$. Let $\|W\|_F \leq M$ with $M$ constant. Define $\widehat{B_{i,t}} = (\Lambda_{t,L}^i)^{-1/2}(U_{t,L}^i)^{\mathsf{T}}B_i(W_t)U_{t,L}^i(\Lambda_{t,L}^i)^{-1/2}$ and $\widehat{C_{i,t}} = (\Lambda_{t,R}^i)^{-1/2}(U_{t,R}^i)^{\mathsf{T}}C_i(W_t)U_{t,R}^i(\Lambda_{t,R}^i)^{-1/2}$ where the $\Lambda_{t,L}^i, U_{t,L}^i, \Lambda_{t,R}^i$ and $U_{t,R}^i$ are the outputs of Algorithm ??. Also let $S_t = \Lambda_{t,L}^{-1/2}U_{t,L}^{\mathsf{T}}G_t U_{t,R}\Lambda_{t,R}^{-1/2}, \kappa_t = \frac{1}{N}\sum_i \lambda_{\min}(\widehat{B_{i,t}})\lambda_{\min}(\widehat{C_{i,t}})$ and $\lambda_{\min}(\Lambda_L), \lambda_{\min}(\Lambda_R) \geq \lambda_0 \geq 0$. Assuming that the projection matrices remain constant during the training. Then for the learning rate $\eta$ and $\min(\kappa_t) > (L_A + 2L_B L_C M^2)$, the Clean satisfies*

$$\|S_t\|_F \leq \frac{\eta}{\lambda_0}(L_A + 2L_B L_C M^2)\|S_{t-1}\|_F + (1 - \frac{\eta\kappa_{t--1}}{\lambda_0})\|S_{t-1}\|_F$$

$$= [1 - \frac{\eta}{\lambda_0}(\kappa_{t-1} - L_A - 2L_B L_C M^2)]\|S_{t-1}\|_F.$$

*Proof.* During the proof we use the Sylvester equality. Let $\otimes$ presents the Kronecker product then for arbitrary matrices $A, B$ and $X$, $\mathrm{vec}(AXB) = (B^{\mathsf{T}} \otimes A)\mathrm{vec}(X)$. By vectorizing the gradient parametric form we have

$$g_t = \mathrm{vec}(G_t) = \mathrm{vec}(\sum_{i=1}^N A_i - \sum_{i=1}^N B_i W_t C_i) = a_t - R_t w_t,$$

where $w_t = \mathrm{vec}(W_t), a_t = \frac{1}{N}\sum_i \mathrm{vec}(A_{i,t})$ and $R_t = \frac{1}{N}\sum_i C_{i,t} \otimes B_{i,t}$. Using the Sylvester equation, the vectorized form of $S_t = \Lambda_{t,L}^{-1/2}U_{t,L}^{\mathsf{T}}G_t U_{t,R}\Lambda_{t,R}^{-1/2}$ is

$$s_t = \mathrm{vec}(\Lambda_{t,L}^{-1/2}U_{t,L}^{\mathsf{T}}G_t U_{t,R}\Lambda_{t,R}^{-1/2}) = (\Lambda_{t,R}^{-1/2}U_{t,R}^{\mathsf{T}}) \otimes (\Lambda_{t,L}^{-1/2}U_{t,L}^{\mathsf{T}})\mathrm{vec}(G_t)$$

$$= (\Lambda_{t,R}^{-1/2}U_{t,R}^{\mathsf{T}}) \otimes (\Lambda_{t,L}^{-1/2}U_{t,L}^{\mathsf{T}})g_t. \tag{15}$$

Moreover, $\tilde{G}_t = U_{t,L}\Lambda_{t,L}^{-1/2}U_{t,L}^{\mathsf{T}}G_t U_{t,R}\Lambda_{t,R}^{-1/2}U_{t,R}^{\mathsf{T}}$ can be written as

$$\tilde{g}_t = \mathrm{vec}(U_{t,L}\Lambda_{t,L}^{-1/2}U_{t,L}^{\mathsf{T}}G_t U_{t,R}\Lambda_{t,R}^{-1/2}U_{t,R}^{\mathsf{T}}) = (U_{t,R} \otimes U_{t,L})(\Lambda_{t,R}^{-1/2}U_{t,R}^{\mathsf{T}}) \otimes (\Lambda_{t,L}^{-1/2}U_{t,L}^{\mathsf{T}})g_t$$

$$= (U_{t,R} \otimes U_{t,L})s_t. \tag{16}$$

Suppose that the Nystrom subspaces remain fixed over a window of iterations. That is, for the left projections we have $U_{t,L} = U_R$ and $\Lambda_{t,L} = \Lambda_L$ and analogously for the right projections. Then $s_t$ and $\tilde{g}_t$ can be restated as:

$$s_t = (\Lambda_R^{-1/2}U_R^{\mathsf{T}}) \otimes (\Lambda_L^{-1/2}U_L^{\mathsf{T}})g_t \quad \text{and} \quad \tilde{g}_t = (U_R \otimes U_L)s_t.$$

Now we derive the recursive form of $g_t$:

$$g_t = a_t - R_t w_t = a_t - R_t w_t - g_{t-1} + g_{t-1}$$
$$= a_t - R_t w_t - a_{t-1} + R_{t-1}w_{t-1} + a_{t-1} - R_{t-1}w_{t-1}$$
$$= a_t - R_t w_t - a_{t-1} + R_{t-1}(w_t - \eta\tilde{g}_t) + a_{t-1} - R_{t-1}w_{t-1}$$
$$= (a_t - a_{t-1}) + (R_{t-1} - R_t)w_t + a_{t-1} - R_{t-1}w_{t-1} - \eta R_{t-1}\tilde{g}_t$$
$$= e_t + g_{t-1} - \eta R_{t-1}\tilde{g}_t$$

where $e_t = (a_t - a_{t-1}) + (R_{t-1} - R_t)w_t$ and we use $w_t = w_{t-1} + \eta g_{t-1}$. By left-multiplying $\Lambda_R^{-1/2}U_R^{\mathsf{T}}) \otimes (\Lambda_L^{-1/2}U_L^{\mathsf{T}})$, we obtain

$$s_t = (\Lambda_R^{-1/2}U_R^{\mathsf{T}}) \otimes (\Lambda_L^{-1/2}U_L^{\mathsf{T}})g_t = (\Lambda_R^{-1/2}U_R^{\mathsf{T}}) \otimes (\Lambda_L^{-1/2}U_L^{\mathsf{T}})(e_t + g_{t-1} - \eta R_{t-1}\tilde{g}_t)$$

$$= (\Lambda_R^{-1/2}U_R^{\mathsf{T}}) \otimes (\Lambda_L^{-1/2}U_L^{\mathsf{T}})e_t + (\Lambda_R^{-1/2}U_R^{\mathsf{T}}) \otimes (\Lambda_L^{-1/2}U_L^{\mathsf{T}})g_{t-1}$$

$$- \eta(\Lambda_R^{-1/2}U_R^{\mathsf{T}}) \otimes (\Lambda_L^{-1/2}U_L^{\mathsf{T}})R_{t-1}\tilde{g}_{t-1}$$

$$= (\Lambda_R^{-1/2}U_R^{\mathsf{T}}) \otimes (\Lambda_L^{-1/2}U_L^{\mathsf{T}})e_t + s_{t-1} - \eta(\Lambda_R^{-1/2}U_R^{\mathsf{T}}) \otimes (\Lambda_L^{-1/2}U_L^{\mathsf{T}})R_{t-1}(U_R \otimes U_L)s_{t-1}$$

$$= (\Lambda_R^{-1/2}U_R^{\mathsf{T}}) \otimes (\Lambda_L^{-1/2}U_L^{\mathsf{T}})e_t + s_{t-1} - \eta(\Lambda_R^{-1/2} \otimes \Lambda_L^{-1/2})(U_R \otimes U_L)^{\mathsf{T}}R_{t-1}(U_R \otimes U_L)s_{t-1}$$

$$\tag{17}$$

Let

$$\hat{R}_t = (U_R \otimes U_L)^\mathsf{T} R_t (U_R \otimes U_L) = \frac{1}{N} \sum_i (U_R \otimes U_L)^\mathsf{T} (C_{i,t} \otimes B_{i,t})(U_R \otimes U_L)$$

$$= \frac{1}{N} \sum_i (U_R^\mathsf{T} C_{i,t} U_R) \otimes (U_L^\mathsf{T} B_{i,t} U_L),$$

then eqn. equation 17 can be presented as

$$s_t = (\Lambda_R^{-1/2} \otimes \Lambda_L^{-1/2})(U_R \otimes U_L)^\mathsf{T} e_t + (I - \eta(\Lambda_R^{-1/2} \otimes \Lambda_L^{-1/2})\hat{R}_{t-1})s_{t-1}. \quad (18)$$

Now we need to bound $s_t$ in above equation. Since $U_L$ and $U_R$ are orthonormal matrices we have $U_L^\mathsf{T} U_L = I$ and $U_R^\mathsf{T} U_R = I$. Therefore by using Sylvester identity and submultiplicavity property of the norm,

$$\left\| (\Lambda_R^{-1/2} \otimes \Lambda_L^{-1/2})(U_R \otimes U_L)^\mathsf{T} e_t \right\|_2 = \left\| \mathrm{vec} \left[ (U_R \otimes U_L)^\mathsf{T} E_t (\Lambda_R^{-1/2} \otimes \Lambda_L^{-1/2}) \right] \right\|_2$$

$$= \left\| (U_R \otimes U_L)^\mathsf{T} E_t (\Lambda_R^{-1/2} \otimes \Lambda_L^{-1/2}) \right\|_F$$

$$\leq \left\| (U_R \otimes U_L)^\mathsf{T} E_t \right\|_2 \left\| \Lambda_R^{-1/2} \otimes \Lambda_L^{-1/2} \right\|_F$$

$$\leq \frac{\|E_t\|_F}{\lambda_0}, \quad (19)$$

where $E_t = \frac{1}{N}\sum_i A_{i,t} - A_{i,t-1} + \frac{1}{N}\sum_i B_{i,t-1} W_t C_{i,t-1} - B_{i,t} W_t C_{i,t}$ is the matrix form of $e_t$. Therefore, we need to bound $\|E_t\|_F$. By Lipchitz-continuity property of $A_i, B_i$ and $C_i$ we have:

$$\|A_t - A_{t-1}\|_F \leq L_A \|W_t - W_{t-1}\|_F = \eta L_A \left\| \tilde{G}_{t-1} \right\|_F \leq \eta L_A \|S_{t-1}\|_F$$

$$\|(B_{t-1} - B_t) W_t C_{t-1}\|_F \leq L_B \|W_t - W_{t-1}\|_F \|W_t\|_F \|C_{t-1}\|_F \leq \eta L_B L_C M^2 \|S_{t-1}\|_F$$

$$\|B_t W_t (C_{t-1} - C_t)\|_F \leq L_C \|W_t - W_{t-1}\|_F \|W_t\|_F \|B_t\|_F \leq \eta L_B L_C M^2 \|S_{t-1}\|_F.$$

Thus the upper bound for $\|E_t\|_F$ can be derived:

$$\frac{\|E_t\|_F}{\lambda_0} \leq \frac{\eta L_A \|S_{t-1}\|_F + \eta L_B L_C M^2 \|S_{t-1}\|_F + \eta L_B L_C M^2 \|S_{t-1}\|_F}{\lambda_0}$$

$$\leq \frac{\eta}{\lambda_0}(L_A + 2L_B L_B M^2) \|S_{t-1}\|_F. \quad (20)$$

To find an upper bound of $\|S_t\|_F$ we also need to find an upper bound for $I - \eta(\Lambda_L^{-1/2} \otimes \Lambda_R^{-1/2})\hat{R}_{t-1}$. This involves finding the minimum eigenvalue of $\hat{R}_t$. Let $\lambda_{i,t} = \lambda_{\min}(U_R^\mathsf{T} C_{i,t} U_R)$ and $\lambda'_{i,t} = \lambda_{min}(U_L^\mathsf{T} B_{i,t} U_L)$, then

$$\lambda_{\min} \left( (\Lambda_L^{-1/2} \otimes \Lambda_R^{-1/2})(U_R^\mathsf{T} C_{i,t} U_R) \otimes (U_L^\mathsf{T} B_{i,t} U_L) \right) =$$

$$\lambda_{\min}(\Lambda_L^{-1/2})\lambda_{\min}(\Lambda_R^{-1/2})\lambda_{\min}(U_L^\mathsf{T} B_{i,t} U_L)\lambda_{\min}(U_R^\mathsf{T} C_{i,t} U_R) \leq \frac{\lambda_{i,t}\lambda'_{i,t}}{\lambda_0}.$$

For any vector $v$ we have

$$v^\mathsf{T} \left[ (\Lambda_L^{-1/2} \otimes \Lambda_R^{-1/2})\hat{R}_t \right] v = \frac{1}{N} \sum_i v^\mathsf{T} \left[ (\Lambda_L^{-1/2} \otimes \Lambda_R^{-1/2})(U_R^\mathsf{T} C_{i,t} U_R) \otimes (U_L^\mathsf{T} B_{i,t} U_L) \right] v$$

$$\geq \frac{1}{N} \sum_i \frac{\lambda_{i,t}\lambda'_{i,t}}{\lambda_0} = \frac{\kappa_t}{\lambda_0}$$

Therefore, $\lambda_{\max}(I - \eta(\Lambda_L^{-1/2} \otimes \Lambda_R^{-1/2})\hat{R}_{t-1}) \leq (1 - \frac{\eta\kappa_{t-1}}{\lambda_0})$. By combining eqn. equation 20 and eqn. equation 18 we obtain:

$$\|S_t\|_F \leq \frac{\eta}{\lambda_0}(L_A + 2L_B L_C M^2) \|S_{t-1}\|_F + (1 - \frac{\eta\kappa_{t-1}}{\lambda_0}) \|S_{t-1}\|_F$$

$$= [1 - \frac{\eta}{\lambda_0}(\kappa_{t-1} - L_A - 2L_B L_C M^2)] \|S_{t-1}\|_F.$$

$\square$

## C.1 RANDOMIZED NYSTRÖM APPROXIMATION

In this section, we give the randomized Nyström approximation algorithms from (Frangella et al., 2023; Tropp et al., 2017)

---

**Algorithm 2** Randomized Nyström Approximation (RandNystromApprox)

---

1: **Input:** Matrix $G \in \mathbb{R}^{m \times n}$, rank size $r \leq m$, $\Omega \in \mathbb{R}^{m \times r}$.

---

2: $\Omega \sim \mathcal{N}(0, I) \in \mathbb{R}^{m \times r}$      {random Gaussian matrix}
3: $\Omega \leftarrow \mathrm{QR}(\Omega, 0)$      {thin QR decomposition}
4: $Y \leftarrow GG^{\mathsf{T}}\Omega$
5: $\nu \leftarrow \mathrm{eps}(\mathrm{norm}(Y, 2))$      {compute shift}
6: $Y_\nu \leftarrow Y + \nu\Omega$      {add shift for stability}
7: $B \leftarrow \Omega^{\mathsf{T}}Y_\nu$
8: $C \leftarrow \mathrm{CHOL}((B + B^{\mathsf{T}})/2)$      {Force symmetry and Cholesky decomposition}
9: $B \leftarrow Y_\nu / C$      {triangular solve}
10: $[U, \Sigma, \sim] = \mathrm{SVD}(B, 0)$      {thin SVD}
11: $\hat{\Lambda} \leftarrow \max\{0, \Sigma^2 - \nu I\}$      {remove shift, compute eigs}
12: **Output:** $[U, \hat{\Lambda}]$

---

# D EXPERIMENTAL SETUP DETAIL

**Fine-tuning GLUE.** As outlined earlier, we evaluated **CLEAN** by fine-tuning RoBERTa-base model on the GLUE benchmark. The list of hyperparameters used in this experiment is also reported in Table 4.

Table 4: Hyperparameters for fine-tuning RoBERTa-base on GLUE.

| | CLEAN | SOAP | LoRA | GaLore | AdamW |
|---|---|---|---|---|---|
| Epochs | | | 3 | | |
| Warm-up | | | $\times$ | | |
| Batch | | | 16 | | |
| Max Len | | | 128 | | |
| Data Type | | | `bfloat32` | | |
| LR | | $\{1e-4, 2e-4, \ldots, 5e-4\}$ / $\{1e-5, \ldots, 5e-5\}$ | | | |
| LR Schedule | | | linear to 0% | | |
| $\beta_1$ | 0.9 | 0.95 | 0.9 | 0.9 | 0.9 |
| $\beta_2$ | 0.999 | 0.95 | 0.999 | 0.999 | 0.999 |
| Weight Decay | $\times$ | $\times$ | $\times$ | $\times$ | $\times$ |
| Dropout | $\times$ | $\times$ | $\times$ | $\times$ | $\times$ |
| Grad Clip | $\times$ | $\times$ | $\times$ | $\times$ | $\times$ |
| Subspace Update Interval $T$ | 32 | 32 | $\times$ | 500 | $\times$ |
| Rank $r$ | 32 | $\times$ | 4/8 | 4/8 | $\times$ |
| Subspace Proj | $\checkmark$ | $\times$ | $\times$ | $\checkmark$ | $\times$ |
| Accum Weight | $\checkmark$ | $\checkmark$ | $\times$ | $\times$ | $\times$ |
| Grad Proj | $\checkmark$ | $\times$ | $\times$ | $\checkmark$ | $\times$ |

**Pre-training on C4.** To illustrate the feasibility of our approach within limited time, we pre-trained the Llama 130M model using the hyperparameters summarized in Table 5.

Table 5: Hyperparameters for training Llama 130M across different optimizers.

| | CLEAN | SOAP | AdamW |
|---|---|---|---|
| Training Steps | | 1300 | |
| Warm-up Steps | | 130 | |
| Max Len | | 1024 | |
| Batch | | 2000 | |
| Token Batch | | 2M | |
| Data Type | | `bfloat16` | |
| LR | | $\{5e-4, 1e-3, 5e-3, 1e-2\}$ | |
| Warm-up Schedule | | linear from 0% | |
| LR Schedule | | cosine to 0% | |
| $\beta_1$ | 0.9 | 0.95 | 0.9 |
| $\beta_2$ | 0.999 | 0.95 | 0.999 |
| $\mu$ | 0.95 | 0.95 | $\times$ |
| Regularization factor $\rho$ | $1e-2$ | $\times$ | $\times$ |
| Weight Decay | $\times$ | $\times$ | $\times$ |
| Dropout | $\times$ | $\times$ | $\times$ |
| Grad Clip | $\times$ | $\times$ | 1.0 |
| Subspace Update Interval $T$ | 10 | 10 | $\times$ |
| Rank $r$ | 256 | $\times$ | $\times$ |
| Subspace Proj | ✓ | ✓ | $\times$ |
| Accum Weight | ✓ | $\times$ | $\times$ |
| Grad Proj | ✓ | ✓ | $\times$ |

## USE OF LLMs

We verify that we use LLMs (e.g., chatGPT, Claude) to polish and rephrase some sentences of the paper text. We also used it for discovery purposes like finding related works.