# OpenReview forum: "Clean: Curvature-Aware Memory-Efficient Optimizer via Nyström Sketching"
_ICLR.cc/2026/Conference — ICLR 2026 Conference Withdrawn Submission_

### Official Review · Reviewer_Dghb · 2025-10-28

**Soundness:** 3
**Presentation:** 3
**Contribution:** 2
**Rating:** 4
**Confidence:** 4

**Summary:**

The paper proposes CLEAN, a memory-efficient curvature-aware optimizer. For each matrix-shaped parameter it maintains low-rank Nyström sketches of the left/right gradient covariances, performs two-sided preconditioning and stores a r1*r2 core. The method tracks Adam-style moments in that core, and transports those moments when the low-rank bases are refreshed. Experiments cover GLUE fine-tuning with RoBERTa-base and a small LLaMA-family 130M pre-training feasibility run. Ablations vary rank r, refresh period T, and smoothing weight μ. The method reports lower optimizer-state memory than baselines (AdamW, SOAP, GaLore, LoRA) with competitive accuracy in fine-tuning and reasonable pre-training curves.

**Strengths:**

1. The paper is well-motivated, and the writings are clear.
2. Convergence analysis is provided and proved in appendix.

**Weaknesses:**

1. The convergence result is proved under a fixed projection (static subspace) at every iteration, but CLEAN’s practical design periodically changes the left/right subspaces and transports moments. Alternatively, I've seen papers adopting a Hamiltonian descent framework that naturally handles dynamic projections and derive stationarity or convergence rate. Essentially, Hamiltonian descent framework is a flow-based descent that models updates within random or evolving subspaces. Moreover, one can prove the convergence with Adam-like optimizer instead of just SGD with Halmiltonian descent.
2. One more problem about this convergence result is that, the current convergence rate could lead to a very strict selection over the learning rate \eta, so that the rate before \|S_{t-1}\|_F is meaningful. Could this issue implicitly suggest the instability of CLEAN?
3. From my perspective, training with gradient projected to subspace is basically a tradeoff between memory, computational complexity and performance. In the paper, memory savings are emphasized, but extra compute from (i) two Nyström sketches (left/right) per refresh, (ii) the SVD/QR truncations used for “concatenate-and-truncate”, and (iii) transport multiplications is not quantified. I suggest also adding some discussion over this. (From table 2, compared with LoRA, CLEAN has lower memory cost, and the performance is also similar, while the training is slower.)
4. For experiments, GLUE with RoBERTa-base (3 epochs) is a small, well-tuned setting where most optimizers perform similarly; the main table shows only marginal gains over low-rank baselines and trails full-rank SOAP/AdamW in places. Moreover, I think results on larger models (e.g., 1–7B) or longer training where the method’s stability over long horizons would better demonstrate the superiority of your method.
5. Currently r1=r2, have you attempted different settings where r1 \neq r2? Or is there any intuition how the training will be regarding subspace gradient projection if we adjust r1, r2 differently? Are they contributing equally to the method? This could essentially be the difference between GaLORE and LoRA. I'm asking this because LoRA+ improves LoRA by using inequivalent learning rates on two low rank matrices.

**Questions:**

see weakness.

---

### Official Review · Reviewer_Ut8f · 2025-10-28

**Soundness:** 3
**Presentation:** 3
**Contribution:** 2
**Rating:** 4
**Confidence:** 3

**Summary:**

This paper aim to address the memory-curvature trade-off in efficient LLMs optimization via the CLEAN, a curvature-aware memory efficient optimizer that approximate the second-order information (the left and right gradient covariances) using randomized Nystrom sketches (Algorithm 1). Specifically, CLEAN approximates the Shampoo preconditioner factors using low-rank randomized Nyström sketches, avoiding computing the full Hessian. Then, the approximated Shampoo preconditioned gradient is used to update the first and second moments of the optimizer.

**Strengths:**

- The authors applied the Nystrom sketch methods enable efficient approximation of Shampoo-style preconditioning in LLMs optimization, reducing the memory footprint of the optimizer states.
- The authors provide illustrative derivation of the proposed method.
- The authors evaluate the performance of CLEAN on various LLMs finetuning tasks.

**Weaknesses:**

**Concern 1. Further experiments on the pretraining setting is needed.**

As this paper aim to reduce the memory cost of the optimizer states, adding more experiments on the pretraining setting would be more convincing in demonstratining the significance of the proposed method. From my perspective, the memory issue is a moderate concern in LoRA finetuning setting, where only a low-rank adapter is trainable.

I recommend the authors to provide more experiments on the C4 pretraining setting, with LLMs with varying sizes and FLOPs budgets against other baselines that exploits low-rank structures of the model parameters / optimizer states.

**Concern 2. The performance gain is marginal.**

As shown in Table 2, the performance gain against the classic LoRA baseline (with fewer ranks) seems to be marginal and inconsistent. The reduction of running time and optimization state memory cost of CLEAN seems to be moderate. Given the complexity of the proposed method, a more evident performance gain would be helpful to demonstrate when this sophisticated method is worthwhile to be applied to real-world settings.

**Concern 3. Runtime efficiency is required.**

Various SVD steps are introduced to achieve curvature-aware update. I wonder to what extent the runtime efficiency is affected by the proposed method, though the memory of optimizer states is reduced.

**Questions:**

See **Weaknesses**.

---

### Official Review · Reviewer_Hb8p · 2025-10-30

**Soundness:** 2
**Presentation:** 2
**Contribution:** 1
**Rating:** 2
**Confidence:** 4

**Summary:**

The authors address the important challenge of efficient optimization of large language models (also applicable to other large models). The proposed method can be thought of as a memory-reduced preconditioner, which can reduce the memory footprint compared with Adam (for example). The main idea is to exploit the low-rank structure of the left and right covariance matrices and approximate them using Nystrom sketching. This means the method can be thought of as SOAP with reduced memory. The method is evaluated on reoberta base and compared to Lora, Galore,  SOAP and AdamW.

**Strengths:**

This is an important problem, with an extensive body of recent research.

Overall, the paper is easy to follow and clear.

**Weaknesses:**

While the problem is interesting, the field is highly competitive, out of the PEFT methods the authors focus their comparison on Lora and Galore. However, several other methods are aligned with this goal and should be discussed and compared to. For example, [1], [2], [3], which actually show gain compared with two-sided preconditioners.

[1] Chen et al. Fira: Can We Achieve Full-rank Training of LLMs Under Low-rank Constraint?

[2] Refael et al. SUMO: Subspace-Aware Moment-Orthogonalization for Accelerating Memory-Efficient LLM Training.

[3] Jordan, Muon.

The results presented in the paper primarily rely on Roberta-base and are evaluated solely on the GLUE benchmark. While I believe that our community should not focus exclusively on acquiring more data and developing more models, I argue that the existing evaluation is too limited for this field and for making conclusions about the method's usability. As authors, you should aim to "sell" your method by demonstrating its applicability to more challenging benchmarks and utilizing up-to-date models.

Even when looking at the existing results, it is not clear what advantage the method provides compared with Lora, or Galore (which uses a much lower rank for the same performance).

**Questions:**

Why are you using SVD and not an efficient randomized scheme such as in [4].
[4] Halko et al. Finding structure with randomness: Probabilistic algorithms for constructing approximate matrix decompositions.

 Two .. in line 43.

How is the time computed for the table, are you taking the SVD into account?

The rank for figure 1 is not low, and the method is still outperformed by the baselines.

You state the need for adaptive rank allocation, which is addressed by [4].
[5] Refael et al. AdaRankGrad: Adaptive Gradient-Rank and Moments for Memory-Efficient LLMs Training and Fine-Tuning

---

### Official Review · Reviewer_vEGF · 2025-10-31

**Soundness:** 2
**Presentation:** 3
**Contribution:** 2
**Rating:** 4
**Confidence:** 3

**Summary:**

The paper introduces CLEAN, a curvature aware, memory efficient optimizer for large models. It combines two sided low rank preconditioning (via Nystrom sketches of left and right gradient covariances) with projection-aware moment transport that realigns optimizer states when the subspace changes. CLEAN targets the trade off between curvature fidelity and memory cost, aiming to approximate balanced second-order preconditioning (as in Shampoo) while keeping the memory growth linear in the number of parameters.

**Strengths:**

1. Memory efficient design: the method achieves a low $O(mr+nr+r^2)$ memory footprint, making it significantly smaller than Adam. The design correctly identifies and attempts to solve known stability issues in low-rank projection optimizers.

2. Empirical stability: CLEAN exhibits low sensitivity to hyperparameters (rank r, refresh interval T, accumulation weight μ) and remains numerically stable, an advantage over GaLore and LDAdam.

**Weaknesses:**

1. Weak empirical results. The paper’s empirical evidence fails to support its claims of effectiveness. The method underperforms standard baselines in both of its target use cases. For pre-training: In the 130M parameter "feasibility study", CLEAN converges to a worse final loss (3.756) than both AdamW (3.604) and SOAP (3.261). For fine-tuning: On GLUE (Table 2), CLEAN's average accuracy (84.90) is substantially lower than the standard AdamW baseline (86.47) and only marginally better than GaLore (84.68), while offering no significant memory advantage over it.

2. The method is a synthesis of existing ideas: two-sided preconditioning (Shampoo), Nystrom approximations, and projection-aware transport (explicitly inspired by LDAdam). This combination is somewhat novel, but its practical utility is not demonstrated by the results.

3. The Convergence theorem (Theorem  4.1) is uninformative and not directly connected to the algorithm’s stochastic dynamics. The assumptions (fixed subspace, parametric gradient model) are unrealistic for LLM training and render the bound vacuous. No rate, or stability analysis under changing subspaces is provided.

**Questions:**

Please refer to the Weaknesses section.

Isn’t Theorem 4.1 valid only for reversible layers [1], as they are characterized by the given parametric gradient form?

How sensitive is CLEAN to the random seed used in the Nystrom projections? Are the orthogonal sketches recomputed at each refresh or fixed throughout training?

[1] YuandongTian, LantaoYu, XinleiChen, andSuryaGanguli. Understandingself-supervised learning
 with dual deep networks, 2021.

---

### Official Review · Reviewer_Ss4v · 2025-11-04

**Soundness:** 1
**Presentation:** 2
**Contribution:** 2
**Rating:** 2
**Confidence:** 5

**Summary:**

This work presents a new memory-efficient optimization algorithm to alleviate the memory cost of optimizer states during training, achieved through low-rank subspace projection. While many variants of low-rank subspace optimizers exist (such as Galore[1], LDAdam[2]), the paper argues that such optimizers are not able to achieve memory efficiency, curvature awareness, and long-horizon stability all at once, and hence proposes a new low-rank subspace optimizer called CLEAN. The proposed algorithm leverages randomized Nystrom approximation to approximate the left and right gradient covariances with low-rank factors, and then leverages those factors for subspace projection akin to Galore but with two-sided projection. The subspace update is performed efficiently by computing rank-$r$ truncated SVD on concatenated subspace factors of the last two computed subspaces (which is itself a rank-$2r$ matrix) to compute the next subspace factors. Moreover, the subspace moment accumulation is done in a subspace-aware manner to account for the fact that gradients are projected into the subspace with evolving subspace factors akin to LDAdam[2]. The memory efficiency of this optimizer is on par with Galore (up to $O(\min(m,n) r)$ improvement). The experimental setup leverages GLUE fine-tuning on RoBERTa-base, and pre-training Llama 130M on C4 dataset. The reported results indicate that CLEAN achieves comparable performance to AdamW and SOAP in both setups, and ablations on the GLUE benchmark show that CLEAN is insensitive to rank and subspace update intervals.

**Strengths:**

1. Memory-efficient training for LLMs is a high-impact direction in optimization research, and this paper focuses on improving the algorithmic design of subspace optimizers, which is indeed a timely and important problem to address.
2. The proposed algorithm (CLEAN) is clearly explained and easy to follow. The use of randomized Nystrom approximation to approximate the left and right gradient covariances, and its subsequent application to subspace projection, is novel. The proposed subspace updates are computationally efficient, novel, and avoid inefficient full-rank matrix SVD computations performed in established subspace optimizers like Galore.

**Weaknesses:**

1. The main goal of this paper is not well motivated and poorly positioned compared to existing literature. I will provide some examples here. The authors mention that "no existing approach simultaneously achieves memory efficiency, curvature awareness, and long-horizon stability (Line 052)." This claim is not entirely accurate. The same way the paper argues the Shampoo optimizer (line 043) is explicitly curvature aware, optimizers like Adam can also be considered curvature aware (Kronecker approximation vs. diagonal approximation of empirical Fisher matrix), and hence regarding Adam as a first-order method (line 040) is not an accurate view of these optimizers. Galore, which is the first subspace variant of AdamW, can therefore be viewed as both curvature aware while maintaining memory efficiency and stability. The one-sided projection in Galore is indeed a design choice over two-sided projection rather than being a limitation as mentioned in line 048. Moreover, in line 111, the authors mention "However, fully storing $L_t$ and $R_t$ still incurs quadratic memory in layer width, which is prohibitive for very wide layers. Our work addresses this limitation by using a low-rank approximation." The issue of wide layers is primarily relevant for embedding layers and the lm head parameters, where practical implementations of Shampoo[4] already handle this with one-sided preconditioning. Moreover, the AdamW versus Shampoo memory comparison is mainly between O(mn) for the second moment versus O(m²+n²), and lines 041-045 should more accurately reflect this. Works like Sketchy[5] maintain a low-rank approximation of left and right gradient covariances to further reduce O(m²+n²) to O(mr+nr).

2. Soundness of the proposed algorithm. The preconditioning (in the context of Shampoo and the equation in line 086 of the paper) and subspace projection are independent concepts in the context of subspace optimizers like Galore. The notion of preconditioning as correctly defined in line 086 of the paper is actually being applied to the subspace update of Galore, which adopts Adam-like preconditioning on the subspace (maintaining element-wise accumulated squares of projected gradients). The proposed CLEAN algorithm seems to also adopt similar subspace preconditioning as Galore (lines 197-198). On the other hand, the factored left and right covariances as leveraged in the CLEAN algorithm do indeed play the role of constructing the subspace projection. Thus, the argument in lines 243-255 (Preconditioning in the low-rank subspace akin to Shampoo) does not accurately reflect this fact. Moreover, the subspace projection is usually done through semi-orthogonal matrices as projection matrices (as done in Galore), and hence the theoretical motivation behind the terms in line 14 of Algorithm 1 needs explanation or justification for having the additional low-rank diagonal matrices multiplied on the left and right sides of the projection.

3. The theoretical analysis provides limited insights: the convergence is based on very strong and unrealistic assumptions (assuming parametric gradient form). While Galore's paper uses the same assumption, the same critique also applies to their convergence result, and this issue is also mentioned in LDAdam, where they introduced a more realistic setting for convergence analysis. Moreover, the derived theorem does not show the benefit of algorithmic components used in this work, nor does it explain the benefits of the proposed algorithm compared to Galore under the same theoretical setup.

4. Experiment rigor:
   - Statistical significance (p-value) is needed for meaningful comparisons on the GLUE benchmark. While this might not be necessary for other setups where optimizers have large gaps in their performance, this is not the case for GLUE benchmarks.
   - The claim of robustness to rank appears to contradict findings from Galore and Dion[3], as the pre-training performance (final validation perplexity) indeed drops noticeably as rank becomes smaller. The robustness claim is being made on the GLUE benchmark, which further strengthens the need for p-values and a careful rank ablation study for the pre-training setup instead. Moreover, the robustness to subspace intervals is also questionable (especially considering the sensitivity of Galore to subspace update interval), and the subspace interval ablation would be more meaningful on the pre-training setup as well.
   - Galore is the key baseline for comparison in this paper, which is currently missing for the pre-training setup.

**Questions:**

1. Could you elaborate more on the theoretical soundness of the accumulation of preconditioners (Line 225-241)? Such accumulation, as shown in Equation 7, would only make sense if the rank of $K_L \in \mathbb{R}^{m \times 2 r_1}$ remains close to $r_1$, but when doing Randomized Nyström, the rank of $K_L$ would actually be expected to be close to $2 r_1$, and hence the error coming from truncating $K_L$ to rank $r_1$ might actually hurt the accumulation purpose. Sketchy[5] maintains a low-rank factorization of left and right gradient covariances, where instead of Nystrom approximation used in this work, they use the Frequent Directions Update to maintain low-rank factors of those covariances. A notion such as escaped mass in lemma 1 of [5] might be helpful to understand my argument.
2. Could you provide p-values for the GLUE fine-tuning setting?
3. Could you add Galore to the pre-training setup and report the final validation loss across ranks and across subspace intervals for both algorithms? Moreover, for fair comparison with Galore, wall-clock time comparison needs to be provided for pre-training as well.



References

[1] Zhao, Jiawei, et al. "Galore: Memory-efficient llm training by gradient low-rank projection." arXiv preprint arXiv:2403.03507 (2024).

[2] Robert, Thomas, et al. "Ldadam: Adaptive optimization from low-dimensional gradient statistics." arXiv preprint arXiv:2410.16103 (2024).

[3] Ahn, Kwangjun, et al. "Dion: Distributed orthonormalized updates." arXiv preprint arXiv:2504.05295 (2025).

[4] Anil, Rohan, et al. "Scalable second order optimization for deep learning." arXiv preprint arXiv:2002.09018 (2020).

[5] Feinberg, Vladimir, et al. "Sketchy: Memory-efficient adaptive regularization with frequent directions." Advances in Neural Information Processing Systems 36 (2023): 75911-75924.

---

### Note · Authors · 2025-12-03

**Comment:**

We would like to thank all reviewers for their constructive comments and valuable feedback. We will consider all of them for our next submission.

**Withdrawal Confirmation:**

I have read and agree with the venue's withdrawal policy on behalf of myself and my co-authors.